# Sepsis-Induced Coagulopathy Phenotype Induced by Oxidized High-Density Lipoprotein Associated with Increased Mortality in Septic-Shock Patients

**DOI:** 10.3390/antiox12030543

**Published:** 2023-02-21

**Authors:** Yolanda Prado, Pablo Tapia, Felipe Eltit, Cristian Reyes-Martínez, Carmen G. Feijóo, Felipe M. Llancalahuen, Claudia A. Riedel, Claudio Cabello-Verrugio, Jimmy Stehberg, Felipe Simon

**Affiliations:** 1Laboratory of Integrative Physiopathology, Faculty of Life Science, Universidad Andres Bello, Santiago 8370186, Chile; 2Millennium Institute on Immunology and Immunotherapy, Santiago 8331150, Chile; 3Unidad de Paciente Crítico Adulto, Hospital Clínico La Florida, La Florida, Santiago 8242238, Chile; 4Department of Urologic Sciences, University of British Columbia, Vancouver, BC V5Z 1M9, Canada; 5Vancouver Prostate Centre, Vancouver, BC V6H 3Z6, Canada; 6Fish Immunology Laboratory, Faculty of Life Sciences, Universidad Andrés Bello, Santiago 8370186, Chile; 7Laboratory of Endocrinology-Immunology, Faculty of Life Sciences, Universidad Andrés Bello, Santiago 8370186, Chile; 8Laboratory of Muscle Pathology, Fragility and Aging, Faculty of Life Science, Universidad Andrés Bello, Santiago 8370186, Chile; 9Center for the Development of Nanoscience and Nanotechnology (CEDENNA), Universidad de Santiago de Chile, Santiago 8350709, Chile; 10Laboratory of Neurobiology, Institute of Biomedical Sciences, Faculty of Medicine and Faculty of Life Science, Universidad Andrés Bello, Santiago 8370186, Chile; 11Millennium Nucleus of Ion Channel-Associated Diseases (MiNICAD), Santiago 8380453, Chile

**Keywords:** oxidized HDL, coagulation, sepsis, coagulopathy, mortality

## Abstract

Sepsis syndrome is a highly lethal uncontrolled response to an infection, which is characterized by sepsis-induced coagulopathy (SIC). High-density lipoprotein (HDL) exhibits antithrombotic activity, regulating coagulation in vascular endothelial cells. Sepsis induces the release of several proinflammatory molecules, including reactive oxygen species, which lead to an increase in oxidative stress in blood vessels. Thus, circulating lipoproteins, such as HDL, are oxidized to oxHDL, which promotes hemostatic dysfunction, acquiring prothrombotic properties linked to the severity of organ failure in septic-shock patients (SSP). However, a rigorous and comprehensive investigation demonstrating that oxHDL is associated with a coagulopathy-associated deleterious outcome of SSP, has not been reported. Thus, we investigated the participation of plasma oxHDL in coagulopathy-associated sepsis pathogenesis and elucidated the underlying molecular mechanism. A prospective study was conducted on 42 patients admitted to intensive care units, (26 SSP and 16 non-SSP) and 39 healthy volunteers. We found that an increased plasma oxHDL level in SSP was associated with a prothrombotic phenotype, increased mortality and elevated risk of death, which predicts mortality in SSP. The underlying mechanism indicates that oxHDL triggers an endothelial protein expression reprogramming of coagulation factors and procoagulant adhesion proteins, to produce a prothrombotic environment, mainly mediated by the endothelial LOX-1 receptor. Our study demonstrates that an increased plasma oxHDL level is associated with coagulopathy in SSP through a mechanism involving the endothelial LOX-1 receptor and endothelial protein expression regulation. Therefore, the plasma oxHDL level plays a role in the molecular mechanism associated with increased mortality in SSP.

## 1. Introduction

Sepsis syndrome is the main cause of mortality in patients admitted to intensive care units (ICU). Sepsis is an uncontrolled response to an infection and is characterized by severe systemic inflammation, which progresses to multiple organ dysfunction syndrome (MODS), which is associated with a high mortality rate [1,2,3]. During sepsis, several critical alterations occur, including a loss of hemostatic control, which promotes a procoagulant phenotype that is a central factor for inducing MODS and increasing mortality [3,4]. In fact, sepsis progression is characterized by disseminated intravascular coagulation (DIC) which generates detrimental effects in organ function and is associated with an increased mortality rate [4,5,6,7,8]. Indeed, rats with DIC induced by endotoxin lipopolysaccharide (LPS) administration, showed a decrease in organ failure after profibrinolytic protein tissue-type plasminogen activator (t-PA) injection and the administration of the anticoagulant heparin-reduced MODS [9,10]. The first diagnostic criteria of DIC were established by the International Society on Thrombosis and Haemostasis (ISTH) [11]. During sepsis, systemic coagulation appears in an early stage of sepsis, which compromises tissue circulation and generates MODS [12]. To obtain an early diagnosis of acute DIC, including sepsis-associated DIC, the Japanese Association for Acute Medicine (JAAM) designed the JAAM-DIC diagnostic criteria [13,14]. Thus, a few years ago, new diagnostic criteria were designed specifically for sepsis-based coagulation disorders, including sepsis-induced coagulopathy (SIC) [15,16].

High-density lipoprotein (HDL) is an accepted vascular protective factor that contributes to hemostatic control by providing protection against vascular diseases [17]. HDL is a lipoprotein that circulates in the bloodstream and exhibits both antithrombotic activity and enhances fibrinolysis, thus playing a significant role in regulating coagulation [18,19]. HDL promotes its protective effects by interacting with several tissues, including endothelial cells (ECs), to trigger several processes to maintain hemostatic balance [20]. ECs control hemostasis equilibrium via the regulation of protein expression, such as coagulation factors and adhesion proteins [21,22].

Sepsis-induced systemic inflammation is characterized by immune system overactivation, which produces several inflammatory molecules, including reactive oxygen species (ROS), generating increased oxidative stress in blood vessels [23,24,25,26,27,28]. This oxidative environment promotes oxidative modifications to several molecules in the plasma. Consequently, circulating lipoproteins, such as HDL, are oxidized through modifications mediated by enzymatic and non-enzymatic oxidative activity to generate oxHDL, the oxidized form of HDL [29,30]. Interestingly, it has been reported that HDL is more susceptible than other lipoproteins to oxidation [31]. It is noteworthy that oxHDL fails to maintain hemostasis control and acquires prothrombotic properties, possibly due to oxHDL-induced procoagulant activity via the impairment of EC functions [32,33]. Some evidence has been reported concerning the detrimental actions of oxHDL during coagulation regulation. The expression of plasminogen activator inhibitor-1 (PAI-1) is stimulated by HDL, which is oxidized by 15-lipoxygenase, causing procoagulant actions [34], suggesting that oxHDL leads to a decrease in fibrinolysis, affecting clot stability. Furthermore, previous evidence has shown that oxHDL plasma levels are inversely associated with D-dimer, fibrinogen plasma, plasmin–α2 plasmin inhibitor complex, and thrombin–anti-thrombin complex levels in diabetic patients [35,36], indicating that oxHDL has an impact on both blood coagulation and fibrinolysis. Thus, during an oxidative-stress-mediated inflammatory condition, native HDL is converted into oxHDL, which is dysfunctional and causes cells to acquire a procoagulant phenotype.

Plasma oxHDL levels are increased in several inflammatory pathological conditions, such as obesity, diabetes, renal failure, cardiovascular disease, rheumatoid arthritis and liver, among several others [37,38,39,40,41]. It has been reported that oxHDL correlates with organ failure severity in septic patients [42]. Congruently, recombinant HDL administration led to a decrease in organ dysfunction in endotoxemic rats [43]. In line with this finding, a decrease in HDL plasma level is a predictor of organ dysfunction in septic patients [44]. However, to the best of our knowledge, whether increased plasma oxHDL participates in hemostatic alterations via EC dysfunction, leading to SIC generation and increased mortality in septic-shock patients (SSP) has not yet been studied. Therefore, we focused on determining whether the plasma oxHDL level in SSP is associated with increased mortality in SSP, and demonstrating that oxHDL induces a procoagulant phenotype mediated by a molecular mechanism that involves changes in the endothelial protein expression associated with both SIC establishment and mortality in SSP.

Our results indicate that circulating oxHDL promotes a procoagulant phenotype, which is strongly associated with SIC and leads to the establishment of increased mortality and an elevated risk of death in SSP. The mechanism underlying this process indicates that circulating oxHDL promotes the endothelial protein expression reprogramming of both coagulation factors and procoagulant adhesion molecules, generating a prothrombotic environment. This process is mediated by the activation of the endothelial lectin-like low-density lipoprotein (LOX-1) receptor and the participation of the endotoxin receptor, Toll-like receptor 4 (TLR-4).

Taken together, the plasma oxHDL level emerges as an important factor involved in increased mortality in SSP.

## 2. Materials and Methods

### 2.1. Patients and Volunteers

The study was conducted in 42 ICU patients, 26 septic-shock patients (SSP) and 16 non-septic-shock patients (NSSP), admitted to ICU at Hospital Clínico Metropolitano La Florida located in Santiago, Chile. The patients were critically ill patients admitted to the ICU (septic and non-septic patients) with a distributive shock. The patients were classified as SSP when infection was detected. NSSP consisted of neurocritical, acute pancreatitis, and post-operative vascular surgery patients. The clinical and demographic features of the patients are provided in Appendix A. The mortality of SSP and NSSP was 46.2% and 50.0%, respectively. As a control group, we recruited 39 healthy volunteers (HV). This study was approved by the local institutional Ethics and Bioethics Review Board (N°141008). Additionally, the Commission of Bioethics and Biosafety of Universidad Andres Bello also approved all experimental protocols (N°002/2020). The investigation conforms with the principles outlined in the Declaration of Helsinki. All participants or their surrogates signed an informed consent form prior to entry into the study.

Inclusion criteria for SSP and NSSP included being aged >18 y.o. without a limitation for resuscitation or suffering from shock defined operationally as a requirement for a norepinephrine (NE) dose of >0.1 g·kg^−1^·min^−1^ to maintain the mean arterial pressure between 65–80 mmHg and a lactate concentration of >4 mmol/L. Furthermore, patients had respiratory support with invasive mechanical ventilation and a C-reactive protein level of ≥15 mg/dL. These criteria had to be met 48 h after patients were admitted to the ICU.

Exclusion criteria for SSP and NSSP comprised the consumption of drugs that modify coagulation, fibrinolysis, and platelet aggregation in the previous 14 d. Additionally, we included solid cancer with a more advanced stage than carcinoma in situ, organ transplantation, leukemia, lymphoma, pregnancy, liver cirrhosis, nephrotic syndrome, chronic dialysis, congestive heart failure and red blood cell transfusion with >2 units within the last 48 h after ICU admission. In addition, SSP and NSSP with chemotherapy, hospitalization, or surgery within the last 3 months prior to ICU admission were also excluded. HV were enrolled from the outpatients in the hospital area and research facilities at Universidad Andres Bello. The abovementioned exclusion criteria were also applied to the HVs. The operational definition of a HV was people without any known chronic disease and explicitly without arterial hypertension, chronic allergic condition, diabetes, a body mass index of >30 kg/m^2^, smoking, pregnancy and coagulation dysfunction. In addition, those subjects with an episode of hospitalization or surgery in the last 3 months prior to enrollment in the study were excluded.

Demographic, clinical and laboratory data were carefully recorded and collected. The Acute Physiology and Chronic Health Evaluation II (APACHE II) score was evaluated after admission to the ICU. The Sequential Organ Failure Assessment (SOFA) score was determined on the day of blood recollection. The management and treatment of patients was carried out by the attending physicians at the ICU without any specific intervention for the purpose of this study. Thirty-day mortality was also recorded.

### 2.2. Plasma HDL and oxHDL Measurements, Extraction, and Determination of HDL Oxidation Kinetics Parameters

Human blood samples were obtained after receiving informed consent from patients and volunteers. Samples were collected in a sodium citrate blood collection Vacutainer^®^ tube. Native HDL and oxidized HDL (oxHDL) were measured using a Human OxHDL ELISA kit from MyBioSource, Inc. (San Diego, CA, USA). HDL fractions were obtained from blood samples by ultracentrifugation using sodium bromide. The HDL fraction was adjusted to 50 μg/mL in a phosphate buffer (10 mmol/L, pH 7.4). Next, 20 μL of 1 mmol/L CuSO_4_ was added and changes in absorbance at 234 nm were recorded to determine the oxidation capacity of lipoproteins. The oxidation kinetics parameters (lag time, slope and maximal oxidation) of oxidized HDL were measured, adapted from previously reported studies [40,45,46]. Lag time denotes the period required to start oxidation, detected by a change in baseline absorbance measured in seconds. Slope represents the oxidation velocity and V*_max_* indicates the maximal oxidation, measured as absorbance/seconds and maximal absorbance/seconds, respectively. Values were obtained by subtracting the baseline absorbance. Kinetic curves were fitted to a sigmoidal curve to obtain slope and maximal oxidation values.

### 2.3. Plasma Measurements of Coagulation Parameters and Secreted Proteins Determination

Human blood samples were obtained after receiving informed consent from patients and volunteers. Samples were collected in a sodium citrate blood collection Vacutainer^®^ tube. Plasma platelet count was measured using a Sysmex k 1000 (Sysmex Inc., Kobe, Japan). Plasma levels of D-dimer and fibrinogen were measured using an ELISA kit (both from R&D Systems, Inc.) in accordance with the manufacturer’s instructions. INR was measured using a proper i-STAT cartridge. Plasma levels of tissue factor (TF), t-PA, tissue-factor pathway inhibitor (TFPI) (all from R&D Systems, Inc.), thrombin-activatable fibrinolysis inhibitor (TAFI), von Willebrand factor (vWF) and p-selectin (P-Sel) (all from Abcam, USA) were measured using ELISA kits, according to the instructions of the manufacturer.

### 2.4. Circulating Endothelial Cell (CECs) Separation and Protein Expression Determination Using Flow Cytometry

CECs, which are composed of circulating endothelial mature cells (CMECs) and circulating endothelial progenitor cells (CEPCs), were separated from blood samples obtained from SSP and NSSP, 48 to 72 h after admission to the ICU, and HV. Blood samples were collected in a 3 ml vacutainer tube containing liquid tripotassium ethylenediaminetetraacetic acid (EDTA) as anticoagulant. The collection of blood samples and the isolation of cells and their analysis were carried out by two personnel who were blinded to patient information, as well as clinical characteristics or the further outcomes of the patients. The CMECs and CEPCs were isolated using magnetic bead-based immunoseparation, as described previously [47,48]. Briefly, after blood samples were obtained, the total mononuclear blood cell fraction was isolated from the blood by Ficoll-Histopaque (Sigma Chemical Co., St. Louis, MO, USA) gradient separation. The mononuclear cell fraction was washed by centrifugation with phosphate-buffered saline solution. Then, the mononuclear blood cell fraction was subjected to immunomagnetic bead capture (IBC) using a bead-conjugated CD133 monoclonal antibody and magnetic cell separation system (Miltenyi Biotec, Bergisch Gladbach, Germany). The captured cells corresponded to an enriched CEPC sample (positive selection, CD133^+^), while the cells contained in the eluted solution contained CMECs (negative selection, CD133^−^). To directly isolate CMECs, eluted fluid was subsequently subjected to a second step of IBC positive selection using a bead-conjugated CD146 monoclonal antibody (Miltenyi Biotec), obtaining an enriched CMEC sample (CD146^+^ and CD133^−^). CMEC and CEPC quantification was performed using flow cytometry. Compensation particles (BD CompBeads) and amine polymer microspheres (Becton Dickinson) were used for compensation [47,48]. Fluorescent-conjugated antibodies against VE-Cadherin^+^ and CD31^+^ and against VEGFR-2^+^ and CD34^+^ were used for the detailed phenotype characterization of CMECs and CEPCs, respectively. Flow cytometry analysis was performed to determine the expression changes in TF, TAFI, t-PA, TFPI, vWF and P-Sel using the corresponding monoclonal antibodies (all from R&D Systems, Inc.), coupled to suitable secondary antibodies conjugated to fluorophores (all from ThermoFisher, Waltham, MA, USA). The labeled cells were then analyzed immediately via flow cytometry (BD FACS Fortessa, BD Biosciences, San José, CA). Color compensation matrices were calculated for each staining combination within each experiment using single-stained antibody. In all analyses, doublets and clusters were eliminated. A minimum of 10,000 events were analyzed.

### 2.5. Endothelial Cell Culture, mRNA Isolation and RT-qPCR

Human aortic endothelial cells (HAEC (Lonza, Chicago, IL) were cultured in EGM-2 medium supplemented with 5% FBS. Experiments were performed in 1% FBS. RT-qPCR experiments were performed to measure TF, TAFI, t-PA, TFPI, vWF and P-Sel mRNA levels in HAEC. Total RNA was extracted with Trizol according to the manufacturer’s protocol (Invitrogen, Carlsbad, CA, USA). DNAse I-treated RNA was used for reverse transcription using the Super Script II Kit (Invitrogen, Carlsbad, CA, USA). Equal amounts of RNA were used as templates in each reaction. Quantitative-PCR was performed using the SYBR Green PCR Master Mix (AB Applied Biosystems, Foster City, CA, USA). Assays were run using a RotorGene instrument (Corbet Research, Sydney, Australia). Data are presented as relative mRNA levels of the gene of interest normalized to relative levels of 28S mRNA and normalized against the control condition.

### 2.6. Generation of Oxidization of HDL

Native HDL (Sigma-Aldrich, St Louis, MO, USA) at a final concentration of 0.5 mg/mL was incubated at 37 °C for 16 h in the presence of 50 μM CuSO_4_ in PBS. The reaction was stopped by storing the oxHDL at 4 °C to prevent further oxidation. The extent of lipoprotein oxidation was monitored by measuring thiobarbituric acid reactive substance (TBARS) formation using the TBARS assay kit (Cayman Chemical Company, Ann Arbor, MI, USA), following the manufacturer’s instructions [49]. For oxHDL, a total of 17.55 ± 1.95 μM MDA was obtained versus 1.12 ± 0.57 μM from native HDL, in a total of seven independent experiments (*p* ≤ 0.001). To chelate copper from the reaction, oxHDL was incubated for 5 min with 100 mg/mL of CHELEX-100 (Bio-Rad Laboratories Inc., Hercules, CA, USA), centrifuged at 4 °C for 1 min at 500× *g*, and the pellet was discarded [50].

### 2.7. Reagents and Inhibitors

The following reagents and inhibitors were used: HDL (Sigma-Aldrich, St Louis, MO, USA), neutralizing anti-LOX antibody (1:50, Abcam) and CLI-095 (Invitrogen, USA). All inhibitors were added 1 h before and maintained throughout the treatment. Buffers and salts were purchased from Merck Biosciences.

### 2.8. Data Analyses

Results are presented as mean ± SD or mean ± 95% confidence interval (CI) for the relative risk. Differences were considered significant at *p* < 0.05. Statistical differences were assessed using the student’s t-test (or Mann–Whitney type), one-way analysis of variance (one-way ANOVA) (or Kruskal–Wallis type) followed by Dunn’s post hoc test and two-way analysis of variance (two-way ANOVA) followed by Tukey post hoc test. See the figure legends for the specific test used. The relationships between variables were assessed by means of correlation analysis using Spearman’s correlation coefficients and linear regression. Survival Kaplan–Meier curves were compared via a log-rank (Mantel-Cox) test and Gehan–Breslow–Wilcoxon test to determine survival rates. Contingency analyses with Fisher’s exact test were used to assess the relative risk of death. The ability of the oxHDL level or INDEX to predict death at 30 days was assessed using the area under the receiver operating characteristic curve (AUROC) with a 95% confidence interval (95% CI). Statistical testing was two-sided and used the 5% significance level. The data were analyzed with GraphPad Prism version 9.4 (GraphPad Software, LLC). Samples used in the study were defined to identify the mean magnitude effect of a >2-fold increase in the oxHDL level between HV and SSP and NSSP with standard deviations of 10%. Accordingly, a sample size of 39 HV, 26 SSP and 16 NSSP, considering the further separation into survival/non-survival and high/low plasma levels or INDEX ratio, would provide a 90% statistical power to detect a >2-fold increase in the oxHDL level using a two-sided 0.05 significance level.

## 3. Results

### 3.1. Increased oxHDL Plasma Level in SSP and NSSP

Blood samples from SSP and NSSP showed an increase in plasma oxHDL levels (Figure 1A), whereas the level of native HDL had decreased (Figure 1B) in comparison to HV. Considering the divergent change between oxHDL and HDL, we determined the oxHDL/HDL INDEX (called INDEX from this point) as a measure of the transition of native HDL to the oxidized form. As shown in Figure 1C, INDEX increased in SSP and NSSP, exhibiting a strong significance in the SSP group

### 3.2. Susceptibility to Oxidation of HDL from SSP

Considering that oxHDL is derived from the oxidation of native HDL, we investigated whether the HDL from patients was more susceptible to oxidation. To that end, the HDL obtained from HV, SSP, and NSSP were subjected to in vitro oxidation to determine the HDL oxidation kinetic parameters, including lag time, oxidation rate, and maximal oxidation. We observed that the lag time of HDL oxidation was similarly shortened in both SSP and NSSP (Figure 1D,E) compared to HV. Interestingly, the HDL oxidation rate measured as the slope of HDL oxidation showed a greater increase in values in SSP rather than NSSP (Figure 1D,F), denoting the susceptibility of HDL from SSP to oxidation. Additionally, the maximal oxidation of HDL in both SSP and NSSP increased, reaching similar maximal values (Figure 1D,G).

### 3.3. Plasma oxHDL Level and INDEX in Non-Survivor SSP

Next, we wondered whether levels of oxHDL and INDEX differentially increased in surviving and non-surviving SSP and NSSP. The results showed that oxHDL levels were higher in non-surviving than surviving SSP, whereas in NSSP, no difference was detected (Figure 2A). Similarly, INDEX was higher in non-surviving than surviving SSP, whereas no change was found in NSSP (Figure 2B). Interestingly, the oxHDL level correlated with a 30-day survival rate in non-surviving SSP, whereas in NSSP, no significant correlation with this survival rate was found (Figure 2C). Such a correlation was stronger (*p* = 0.0012 versus 0.0125) when INDEX was analyzed (Figure 2D), suggesting that INDEX correlates better than oxHDL level with the 30-day survival rates of non-surviving patients in the SSP group.

Considering that a higher oxHDL level could be more deleterious than lower ones, we analyzed the survival curves of SSP and NSSP grouped into high- and low-oxHDL levels. High- and low-oxHDL groups were determined using the median concentration depicted in Figure 1A as the threshold. The results showed a significant difference in the high-oxHDL group compared with the low-oxHDL group in SSP patients, as indicated by the log-rank (Mantel–Cox) test as shown in Figure 2E. To give more weight to deaths at early time points, the Gehan–Breslow–Wilcoxon test was used, and results showed that the high-oxHDL group had an increase in death incidence compared to low-oxHDL (Figure 2E). However, NSSP showed no differences between high- and low-oxHDL levels in the different groups of patients (Figure 2F). Next, whether a higher INDEX value reflected more deleterious conditions was assessed. To that end, high- and low-INDEX groups were determined using the median value depicted in Figure 1C as the threshold. Similarly, the high-INDEX survival curve showed significant differences compared to the low-INDEX group, as indicated by the log-rank (Mantel–Cox) test and Gehan–Breslow–Wilcoxon test, as shown in Figure 2G. In addition, NSSP showed no differences between the high- and low-INDEX levels in different groups of patients (Figure 2H).

### 3.4. High-INDEX Increases Relative Risk of Death in SSP

After considering the results showed above, we performed a contingency analysis to determine the relative risk of death between high- and low- oxHDL and INDEX groups in SSP and NSSP. The results showed that high-oxHDL exhibited an increased, although not significant, risk of death compared to low-oxHDL (Figure 2I). Similarly, NSSP showed no difference in the relative risk of death (Figure 2J). It is noteworthy that the results showed that high-INDEX SSP exhibited a significant increase in the risk of death compared to the low-INDEX group (Figure 2K), whereas NSSP showed no difference in relative risk of death (Figure 2L).

Taking the results in Figure 2 together, an association between increased oxHDL levels and INDEX with SSP but not NSSP is suggested. It is noteworthy that the INDEX shows a stronger potential than the oxHDL level for predicting mortality and risk of death in SSP.

### 3.5. High-oxHDL Level and High-INDEX Correlates with SIC Score in SSP

After considering the prominent role of the plasma oxHDL level and INDEX in SSP, but not in NSSP, we focused our investigation on SSP. Thus, we investigated whether the oxHDL level and INDEX are linked to coagulation parameters to generate a procoagulant phenotype in SSP. As shown in Figure 3A, the platelet count was reduced in SSP compared to healthy volunteers, a finding that is consistent with a platelet consumption phenotype. Interestingly, the high-oxHDL and high-INDEX groups of patients showed a reduced platelet count (Figure 3B) compared to the low-oxHDL and low-INDEX groups, respectively.

Correlation analyses showed that the high-oxHDL and high-INDEX groups correlate with platelet count (Figure 3C, upper panels), whereas low-oxHDL and low-INDEX (Figure 3C, lower panels) showed no significant correlations. D-dimer serves as a marker of coagulation, since it is a fibrin degradation product that increases as a result of fibrinolysis [45,51]. Our results demonstrated that SSP showed an increase in the plasma levels of D-dimer (Figure 3D). The high-oxHDL and high-INDEX groups of patients exhibited an increased plasma D-dimer level (Figure 3E). Additionally, correlation analyses showed that the high-oxHDL and high-INDEX groups correlate with D-dimer levels (Figure 3F, upper panels), whereas low-oxHDL and low-INDEX (Figure 3F, lower panels) showed no such correlation. Fibrinogen was used as a coagulation marker because it decreases as consequence of conversion to fibrin. SSP showed a decreased plasma fibrinogen level (Figure 3G). The high-oxHDL and high-INDEX groups of patients displayed a decreased plasma fibrinogen level (Figure 3H). A decreased plasma fibrinogen level correlated with the high-INDEX group (Figure 3I, upper-right panel). The high-oxHDL, low-oxHDL, and low-INDEX groups did not correlate with plasma fibrinogen levels (Figure 3I, upper-left and lower panels). Because prothrombin time (PT) is prolonged in the procoagulant phenotype, we measured this parameter. To standardize PT measurements, the international normalized ratio (INR) was chosen. The results indicate that SSP showed an increase in INR (Figure 3J). The high-oxHDL and high-INDEX groups of patients showed increased INR (Figure 3K). The correlation analyses showed that the high-oxHDL and high-INDEX groups correlated with INR (Figure 3L, upper panels), while low-oxHDL and low-INDEX (Figure 3L, lower panels) showed no correlations. We also measured the activated partial thromboplastin time (aPPT) since it is prolonged in coagulopathy during sepsis. The results indicate that SSP exhibited a prolonged aPPT (Appendix A), a finding that correlated with the high-INDEX group, whereas the high-oxHDL, low-oxHDL and low-INDEX groups did not correlate with plasma aPPT levels (Appendix A).

A plasma determination of platelet count, D-dimer, fibrinogen, and INR are frequently used to determine ISTH-DIC incidence based on DIC score calculations, whereas only platelet count, INR, and SOFA are needed for SIC determination. Thus, we determined the DIC score to evaluate its association with oxHDL and INDEX levels. To accomplish this task, the ISTH-DIC and SIC score algorithms were used [15,52,53] to determine coagulopathy occurrences as an ISTH-DIC score of ≥5 or SIC score of ≥4. The results showed that the calculated ISTH-DIC score from SSP patients ranged between 2 and 7. The high-oxHDL and high-INDEX groups of patients showed an ISTH-DIC score (Figure 3M,P upper panels, respectively). Correlation analyses showed that the high-oxHDL (Figure 3N upper panel) and high-INDEX (Figure 3Q upper panel) groups correlated with the calculated DIC scores, whereas low-oxHDL and low-INDEX (Figure 3O,R, upper panels, respectively) groups did not show significant correlations. Similarly, the high-oxHDL and high-INDEX groups showed correlating SIC scores (Figure 3M,P lower panels, respectively). Correlation analyses showed that the high-oxHDL (Figure 3N lower panel) and high-INDEX groups (Figure 3Q lower panel) correlated with the calculated SIC scores, whereas the low-oxHDL and low-INDEX groups (Figure 3O,R, lower panels, respectively) did not show significant correlations. These results showed similar results between ISTH-DIC and SIC scores, but the SIC score showed a major ability to detect a major proportion of patients undergoing SIC (score ≥ 4).

After taking into account that plasma oxHDL levels lead to the induction of a procoagulant phenotype and associate with a ≥4 SIC score, we determined whether oxHDL led to an increase in coagulation in blood vessels. A zebrafish model was used, which allowed us to observe in vivo coagulation in the vasculature due the vasculature’s transparency. To that end, 4 dpf wild-type zebrafish larvae were microinjected with saline solution, HDL, oxHDL, or not injected (control), and 24 h later, thrombus formation was analyzed by means of o-dianisidine staining in the caudal vein (Appendix A). The results showed that zebrafish larvae treated with oxHDL exhibited significant coagulation in the posterior part of the caudal vein compared to those treated with HDL, saline solution, or not treated (Appendix A). In addition, we determined if the existence of coagulation was coupled to changes in the blood flow. Thus, 4 dpf Tg(fli1:eGFP)y^1^ larvae, that have the vasculature and platelets fluorescently green labeled, were microinjected with saline solution, HDL, oxHDL, or not injected (control). After 24h, an in vivo determination of the number of platelets observed in 60 seconds via time-lapse analysis in a section of the caudal vein was performed (Appendix A). The results showed that zebrafish larvae treated with oxHDL exhibited a significant reduction in the platelet flow compared to those treated with HDL, saline solution, or not treated (Appendix A).

### 3.6. Plasma oxHDL Level and INDEX Are Associated with Increased Mortality in SSP

To evaluate the capacity to predict mortality through measuring the plasma oxHDL level and INDEX from SSP, we used an area under the receiver operating characteristic curve (AUROC) analysis. The plasma oxHDL level and INDEX from SSP showed a high predictive capacity (Figure 4) showing AUROC values superior to the diagonal nondiscrimination line, which denotes their statistical significance. The AUROC analysis performed in all SSP showed that the oxHDL level is a better predictor than INDEX (Figure 4A). Interestingly, when AUROC analysis was performed in high-oxHDL and high-INDEX SSP groups, INDEX was a better predictive marker than oxHDL level (Figure 4B). Notably, in both cases, the capacity to predict mortality outcomes based on oxHDL and INDEX was better when compared with the APACHE-II and SOFA scores, which are scores of severity and prognosis, respectively, and are widely used for critical patient evaluations (Figure 4A,B).

These findings indicate that plasma oxHDL levels and particularly the INDEX are associated with increased mortality in SSP.

### 3.7. Plasma oxHDL Level and INDEX Correlate with Coagulation Factors and Platelet Adhesion Proteins Level in SSP

Considering both the plasma oxHDL level and INDEX association between an increased oxHDL level and pro-coagulant coagulopathy parameters, the SIC score shown in Figure 3 and the finding that oxHDL treatment induced vascular coagulation in the zebrafish model (Appendix A), we investigated the underlying mechanisms involved in oxHDL-induced coagulation. To that end, we detected several circulating coagulation factors and adhesion proteins involved in coagulation. SSP plasma showed increased prothrombotic TF levels, as shown in Figure 5A. Importantly, the high-oxHDL and high-INDEX groups of patients showed an increased TF plasma level (Figure 5B), compared to the low-oxHDL and low-INDEX groups, respectively. Correlation analyses showed that the high-oxHDL and high-INDEX groups correlated with TF level (Figure 5C, left panels). Interestingly, the SIC score from the high-oxHDL and high-INDEX patients were found to correlate with the TF level (Figure 5C, right panels). Additionally, plasma samples from SSP showed an increase in the levels of the antifibrinolytic factor TAFI, as shown in Figure 5D. Notably, the high-oxHDL and high-INDEX groups of patients exhibited an increased TAFI plasma level (Figure 5E).

Correlation analyses showed that the high-INDEX group and SIC scores from high-INDEX patients correlated with TF levels (Figure 5F, lower panels). The high-oxHDL and SIC score from high-oxHDL showed no significant correlation (Figure 5F, upper panels). Furthermore, the t-PA decreased in SSP (Figure 5G). The high-oxHDL and high-INDEX groups of patients showed a decreased t-PA plasma level (Figure 5H). Correlation analyses indicated that the high-INDEX group, and SIC score from the high-INDEX patients correlated with the t-PA level (Figure 5I, lower panels). The high-oxHDL and SIC scores from high-oxHDL groups showed no significant correlations (Figure 5I, upper panels). Furthermore, the antithrombotic molecule TFPI, as shown in Figure 5J, showed a decrease in plasma levels in SSP. The high-oxHDL and high-INDEX groups of patients displayed a decreased TFPI plasma level (Figure 5K). Correlation analyses showed that the high-INDEX group correlated with the TFPI level (Figure 5L, left-lower panel). Similarly, the SIC score from high-oxHDL and high-INDEX patients correlated with TFPI levels (Figure 5L, right panels). The high-oxHDL group showed no significant correlation (Figure 5L, left-upper panels). On the other hand, the plasma levels of soluble vWF and soluble p-selectin (svWF and sP-Sel, respectively), both of which promote platelet adhesion to the endothelium, were also evaluated. SSP plasma showed increased levels of svWF (Figure 5M), and the high-oxHDL and high-INDEX groups of patients displayed an increased svWF plasma level (Figure 5N). Correlation analyses showed that the high-oxHDL and high-INDEX groups correlated with the svWF level (Figure 5O, left panels). The SIC score from the high-oxHDL and high-INDEX groups of patients also correlated with the svWF level (Figure 5O, right panels). SSP plasma showed an increase in levels of sP-Sel (Figure 5P). The high-oxHDL and high-INDEX groups of patients showed an increased sP-Sel plasma level (Figure 5Q). Correlation analyses showed that the high-INDEX group (Figure 5R, left-lower panel), and SIC score from the high-oxHDL and high-INDEX patients were found to correlate with the sP-Sel level (Figure 5R, right panels). The high-oxHDL group showed no significant correlation with sP-Sel (Figure 5R, left-upper panel). The ISTH-DIC score showed similar results as those shown for the SIC score (Appendix A). The results obtained from the low-oxHDL and low-INDEX groups of patients showed no significant correlations (not shown).

These data indicate that altered levels of coagulation factors and platelet adhesion proteins in SSP appear to be associated with the high-oxHDL and high-INDEX patients, and with the final event of coagulation, since an association with the calculated SIC scores in the high-oxHDL and high-INDEX groups was found.

### 3.8. Circulating Endothelial Cells from SSP Exhibited Modified Coagulation Factors and Platelet Adhesion Protein Expression, which Correlate with Plasma oxHDL and INDEX

Considering that the vascular endothelium regulates hemostasis via both coagulation factor production and platelet adhesion protein generation and that oxHDL induces procoagulant actions via the impairment of EC function, we investigated whether ECs from SSP showed these features. It has been reported that septic patients showed increased CECs, which are composed from CMECs, which are a suitable model for studying vascular ECs from patients, and also from CEPCs [47,48,54]. Thus, analyses were performed on CECs obtained from the high- and low-oxHDL groups and compared with those from healthy volunteers. Positive and negative magnetic bead-based immunoseparation was performed to successfully separate the CMECs (CD146^+^, CD133^−^) and CEPCs (CD133^+^) by means of appropriate markers, after which cells were subject to flow cytometry analyses to determine changes in the expression of coagulation factors and platelet adhesion proteins (Appendix A). The CMECs from SSP showed an increase in TF expression (Figure 6A). The high-oxHDL and high-INDEX groups of patients showed increased TF expression (Figure 6B). Correlation analyses showed that the high-INDEX group correlates with TF expression (Figure 6C, left-lower panel). Similarly, the SIC score from the high-oxHDL and high-INDEX patients were found to correlate with TF expression (Figure 6C, left panels). The high-oxHDL group showed no significant correlation (Figure 6C, left-upper panel). CMECs from SSP showed an increase in TAFI expression (Figure 6D). The high-oxHDL and high-INDEX groups of patients exhibited an increased TAFI expression (Figure 6E). Correlation analyses showed that the high-INDEX group and SIC score from the high-INDEX patients were found to correlate with TAFI expression (Figure 6F, lower panels), whereas high-oxHDL and the SIC score from high-oxHDL showed no significant correlation with the SIC score (Figure 6F, upper panels). CMECs from SSP showed a decreased t-PA expression (Figure 6G). The high-oxHDL and high-INDEX groups of patients displayed a decreased t-PA expression (Figure 6H). Correlation analyses showed that high-INDEX and SIC score from high-INDEX groups correlated with t-PA expression (Figure 6I, lower panels), whereas high-oxHDL and the SIC score from the high-oxHDL groups showed no significant correlation (Figure 6I, upper panels). CMECs from SSP showed a decrease in TFPI expression (Figure 6J). The high-oxHDL and high-INDEX groups of patients showed a decreased TFPI expression (Figure 6K). Correlation analyses showed that the high-INDEX group and SIC score from the high-INDEX group of patients correlated with TFPI expression (Figure 6L, lower panel) whereas high-oxHDL and SIC score from high-oxHDL showed no significant correlation (Figure 6L, upper panel). CMECs from SSP showed an increase in vWF expression (Figure 6M). High-oxHDL and the high-INDEX groups of patients showed increased vWF expression (Figure 6N). Correlation analyses showed that high-oxHDL and high-INDEX and the SIC score from the high-oxHDL and high-INDEX groups of patients correlated with vWF expression (Figure 6O). CMECs from SSP showed an increase in P-Sel expression (Figure 6P). The high-oxHDL and high-INDEX groups of patients showed an increased P-Sel expression (Figure 6Q). Correlation analyses showed that high-oxHDL and high-INDEX (Figure 6R, left panels) and the SIC score from the high-INDEX group (Figure 6R, right-lower panel) of patients were found to correlate with P-Sel expression, whereas the SIC score from the high-oxHDL patients showed no significant correlation (Figure 6R, right-upper panel). Similar results were obtained after testing CEPCs (not shown). The ISTH-DIC score showed similar results as shown for the SIC score (Appendix A). The results obtained from the low-oxHDL and low-INDEX groups of patients showed no significant correlation with P-Sel (not shown).

These results indicate that ECs from SSP express altered levels of coagulation factors and platelet adhesion proteins, both of which are associated with a high-oxHDL and high-INDEX level in addition with the calculated SIC score in the high-oxHDL and high-INDEX groups.

### 3.9. Endothelial Cells Exposed to Plasma from SSP Modified Coagulation Factors and Platelet Adhesion Protein Expression, Which Correlate with Plasma oxHDL and INDEX

After considering the previously mentioned results, we tested whether cultured ECs exposed to plasma extracted from SSP showed similar changes to those observed in CECs. To that end, plasma from high-oxHDL and low-oxHDL patient groups were extracted and added to cultured EC, after which the expression of coagulation factors and platelet adhesion proteins were measured (Appendix A).

Cultured ECs exposed to plasma obtained from the high-oxHDL patients showed an increase in TF mRNA expression (Figure 7A). The high-oxHDL and high-INDEX groups of patients showed an increased TF mRNA expression (Figure 7B). Correlation analyses showed that high-oxHDL and high-INDEX (Figure 7C, left panels) and the SIC score from the high-INDEX (Figure 7C, right-lower panels) group of patients correlated with TF mRNA expression, whereas the SIC score from the high-oxHDL patients showed no significant correlation with TF mRNA expression (Figure 7C, right-upper panel). Cultured ECs exposed to plasma obtained from the high-oxHDL patients showed an increase in TAFI mRNA expression (Figure 7D). The high-oxHDL and high-INDEX groups of patients showed an increased TAFI mRNA expression (Figure 7E). Correlation analyses showed that high-oxHDL and high-INDEX (Figure 7F, left panels) and the SIC score from the high-INDEX (Figure 7F, right-lower panels) group of patients correlated with TAFI mRNA expression, whereas the SIC score from the high-oxHDL patients showed no significant correlation with TAFI mRNA expression (Figure 7F, right-upper panel). Cultured ECs exposed to plasma obtained from high-oxHDL patients showed a decrease in t-PA mRNA expression (Figure 7G). The high-oxHDL and high-INDEX groups of patients showed a decreased t-PA mRNA expression (Figure 7H). Correlation analyses showed that high-oxHDL and high-INDEX (Figure 7I, left panels) and the SIC score from the high-INDEX (Figure 7I, right-lower panels) group of patients correlated with t-PA mRNA expression, whereas the SIC score from high-oxHDL patients showed no significant correlation (Figure 7I, right-upper panel). Cultured ECs exposed to plasma obtained from the high-oxHDL patients showed a decrease in TFPI mRNA expression (Figure 7J). The high-oxHDL and high-INDEX groups of patients showed a decreased TFPI mRNA expression (Figure 7K). Correlation analyses showed that high-oxHDL and high-INDEX and the SIC score from the high-oxHDL and high-INDEX groups of patients correlated with TFPI mRNA expression (Figure 7L). Regarding vWF and P-Sel adhesion protein expression levels, the results showed that cultured ECs exposed to plasma obtained from high-oxHDL patients showed an increase in vWF mRNA expression (Figure 7M). The high-oxHDL and high-INDEX groups of patients showed an increased vWF mRNA expression (Figure 7N). Correlation analyses showed that high-INDEX and the SIC score from the high-INDEX groups were found to correlate with vWF mRNA expression (Figure 7O, lower panels), whereas high-oxHDL and the SIC score from the high-oxHDL patients showed no significant correlation (Figure 7O, upper panels). Cultured ECs exposed to plasma obtained from the high-oxHDL patients showed an increase in P-Sel mRNA expression (Figure 7P). The high-oxHDL and high-INDEX groups of patients showed increased P-Sel mRNA expression (Figure 7Q). Correlation analyses showed that high-INDEX (Figure 7R, left-lower panel) and the SIC score from the high-oxHDL and high-INDEX groups of patients correlated with P-Sel mRNA expression (Figure 7R, right panels), whereas the high-oxHDL group did not show correlation (Figure 7R, left-upper panel). The ISTH-DIC score showed similar results as shown for the SIC score (Appendix A). The results obtained from the low-oxHDL and low-INDEX groups of patients showed no significant correlation with this parameter (not shown).

### 3.10. LOX-1 and TLR-4 Receptors Mediate Actions of SSP Plasma in Endothelial Cells

Although patients’ plasma contains lipoproteins, including HDL and oxHDL, several other molecules are also present. Considering that bacterial infection is the main difference between SSP and NSSP, circulating bacteria and bacterial endotoxin play a role in SSP pathogenesis. LPS is the main circulating endotoxin detected in SSP infected with Gram-negative bacteria.

Therefore, oxHDL and endotoxin are concurrently present in SPP plasma, and both elicit actions in tissues, including the endothelium. The actions of oxHDL and endotoxin in ECs are mediated by their endothelial receptors, LOX-1 and TLR-4, respectively [27,55,56]. Thus, we investigated the role played by these endothelial receptors in the actions of SSP plasma in terms of modifying coagulation factors and platelet adhesion protein expression. To that end, ECs exposed to plasma from high-oxHDL and high-INDEX patient groups were treated with a neutralizing antibody against LOX-1 and the TLR-4 specific inhibitor, CLI-095, after which mRNA expression of coagulation factors and platelet adhesion proteins were measured.

The results showed that both LOX-1 and TLR-4 inhibition caused a partial decline in the increase in TF mRNA expression (Figure 8A) and TAFI (Figure 8B) when ECs were exposed to high-oxHDL (Figure 8A,B, left panels) and high-INDEX (Figure 8A,B, right panels). Furthermore, LOX-1 inhibition led to a strong reduction in the decrease in mRNA expression of t-PA and TFPI (Figure 8C,D, respectively) in ECs that were exposed to high-oxHDL (Figure 8C,D, left panels) and high-INDEX (Figure 8C,D, right panels), whereas TLR-4 inhibition provoked a slight lessening in the increase in mRNA expression of t-PA and TFPI (Figure 8C,D, respectively). In terms of adhesion protein expression, the results showed that LOX-1 inhibition led to a strong reduction in the increase in mRNA expression of vWF, whereas TLR-4 inhibition provoked a minor decrease (Figure 8E) in ECs that were exposed to high-oxHDL (Figure 8E, left panel) and high-INDEX (Figure 8E, right panel). In the case of P-Sel, the results showed that both LOX-1 and TLR-4 inhibition caused a partial reduction in the increase in mRNA expression of P-Sel (Figure 8F) in ECs that were exposed to high-oxHDL (Figure 8F, left panel) and high-INDEX (Figure 8F, right panel).

These results indicate that oxHDL via activation of its receptor, LOX-1, lead to the promotion of an endothelial procoagulant phenotype by controlling the expression of t-PA, TFPI and vWF, whereas that the activation of the endotoxin receptor, TRL-4, showed only a slight tendency toward such a phenotype. Furthermore, TF, TAFI and P-Sel expression levels are controlled by the concomitant actions of oxHDL and endotoxin.

### 3.11. Endothelial Cells Exposed to Exogenous SSP Plasma Preparation Modified Coagulation Factors and Platelet Adhesion Protein Expression

Considering that SSP plasma elicit a prothrombotic phenotype in ECs, we hypothesized that cultured ECs exposed to an exogenous preparation of plasma-containing oxHDL and endotoxin, which mimics SSP plasma, would exhibit similar effects in protein expression profiles versus that observed in CECs from SSP as shown in Figure 6 and in cultured ECs exposed to SSP plasma as shown in Figure 7.

To that end, exogenous plasma samples were prepared by combining oxHDL and endotoxin. The exogenous plasma preparation of SSP high- and low-oxHDL was designed using the median concentration of oxHDL measured in plasma obtained from high- and low-oxHDL groups (SSP-oxHDL^HIGH^+Endo and SSP-oxHDL^LOW^+Endo, respectively). In addition, exogenous plasma preparations of SSP high- and low-INDEX were designed using the concentration of oxHDL and HDL measured in the median INDEX value determined in plasma obtained from high- and low-INDEX groups (SSP-INDEX^HIGH^+Endo and SSP-INDEX^LOW^+Endo, respectively). Endotoxin was added at a concentration used in a previous study by us for similar in vitro studies in ECs [57,58]. Additionally, the exogenous plasma preparation from the HV was designed by combining a suitable amount of HDL and oxHDL, but without endotoxin (HV-oxHDL and HV-INDEX), as shown in Appendix A.

The results indicated that ECs exposed to an exogenous preparation of SSP-oxHDL^HIGH^+Endo and SSP-INDEX^HIGH^+Endo elicited an increase in TF and TAFI mRNA expression (Figure 9A and B, respectively). Furthermore, ECs exposed to SSP-oxHDL^HIGH^+Endo and SSP-INDEX^HIGH^+Endo showed a decrease in the expression of t-PA and TFPI mRNA expression (Figure 9C and D, respectively). In addition, ECs exposed to SSP-oxHDL^HIGH^+Endo and SSP-INDEX^HIGH^+Endo showed an increase in the mRNA expression of adhesion proteins vWF and P-Sel (Figure 9E,F, respectively). The treatment with exogenous SSP-oxHDL^LOW^+Endo and SSP-INDEX^LOW^+Endo yielded no significant effects compared to HV-oxHDL and HV-INDEX, respectively (not shown).

Taken together, these results indicate that SSP produces a procoagulant phenotype by modulating plasma coagulation factors and platelet adhesion protein levels. It is noteworthy that the oxHDL level and, principally, INDEX are strongly associated with the loss of hemostasis in SSP. The results indicated that oxHDL promotes procoagulant activity by means of changing the endothelial protein expression profile, mainly through LOX-1 endothelial receptors, and supported by TLR4. Notably, these findings indicate that the plasma oxHDL level and, mainly, its INDEX are associated with coagulopathy in SSP.

## 4. Discussion

The results from this study indicate that SSP and NSSP showed increases in both oxHDL and INDEX and decreases in HDL plasma levels. The HDL from SSP is more susceptible to oxidation than NSSP HDL, a finding that depicts an oxidation-mediated conversion of the HDL pool into oxHDL. Furthermore, the oxHDL level and INDEX correlate in non-surviving SSP but not in NSSP. Interestingly, the high-oxHDL level and high-INDEX groups of SSP showed increases in mortality and risk of death. In addition, the high-oxHDL level and high-INDEX groups of patients were found to be associated with higher ISTH-DIC and SIC scores in SSP. Notably, a plasma high-oxHDL level and high-INDEX showed significant AUROC analyses for predicting mortality in SSP, and both parameters were better for such a prediction than APACHE II and SOFA scores. The underlying mechanism indicates that oxHDL promotes an endothelial reprogramming in protein expression patterns. OxHDL-treated ECs and high oxHDL SSP samples show an increase in the expression and secretion of the procoagulant factors, TF and TAFI, as well as decreases in the anticoagulant molecules t-PA and TFPI, thus generating a prothrombotic environment. Additionally, the expression and secretion of the procoagulant adhesion proteins, vWF and P-Sel, were increased. Finally, the oxHDL-induced endothelial protein expression change was found to require the activation of the endothelial LOX-1 receptor, which is supported, but not required, by the action of the endotoxin receptor TLR-4.

The role of HDL as an antithrombotic factor has been widely demonstrated [17,18,19,20]. However, it is well known that during sepsis, a large amount of ROS is generated and interacts with HDL to generate oxHDL, which exhibits dysfunctional functions directly involved in pathophysiological processes [32,33]. Increased oxHDL and decreased HDL plasma levels in SSP are involved in other inflammatory diseases, including obesity and type 2 diabetes mellitus [35,36,59,60]. Additionally, increased levels of HDL have been associated with endothelial dysfunction [61,62,63].

After taking into account the results presented in this study, it is reasonable to hypothesize that the excess of HDL is converted into dysfunctional HDL, after which it promotes endothelial damage. Of note is that HDL from SSP is more susceptible to oxidation than is NSSP. This feature could be explained as based on the underlying infection-induced oxidative stress in SSP, which is absent in NSSP. This idea is supported by the notion that LDL oxidation is influenced by infection [64,65]. In fact, endothelial infections caused by the bacterium Chlamydia pneumoniae promote LDL oxidation [65]. Interestingly, the oxidation of HDL appears to be influenced further by serum lipids from the non-HDL fraction, which includes LDL and VLDL [66]. Similarly, in patients with renal failure, less resistance of LDL to oxidation related to the decrease in HDL was observed, suggesting that LDL oxidation depends on the presence of further lipids circulating in blood [67]. Additionally, structural differences between distinct HDL subclasses have no impact on their susceptibilities to oxidation [68], reinforcing the idea that changes in all lipoprotein levels circulating in the bloodstream could influence HDL oxidation. Importantly, it has been reported that the susceptibility of HDL oxidation is associated with age, after observing an increased capacity for HDL oxidation that is associated with aging, a process that has a crucial impact on thrombotic alteration in the elderly population [69].

The days of hospitalization at the ICU of non-surviving SSP correlated with the plasma oxHDL level, indicating that higher oxHDL levels are associated with mortality. Notably, the higher portion of oxHDL levels showed an increase in both mortality and risk of death compared to the lower half. This finding indicates that although all SSP exhibits increased oxHDL compared to HV, a mortality increase is linked only to higher oxHDL values. It is noteworthy that the high-INDEX value strongly associates with mortality and risk of death. Indeed, a high-INDEX value displayed a stronger association with several of the coagulation parameters evaluated throughout the experiments in this study. This finding agrees with those proposed by Guirgis et al., in which dysfunctional HDL (Dys-HDL) was found to be useful as a biomarker for early sepsis [42]. In contrast, in this study, it was hypothesized that early Dys-HDL levels would correlate with SOFA scores in the first 48 h, whereas in the present study, we showed a low correlation between both oxHDL level and INDEX values, and SOFA scores. Further experiments could be performed to understand this difference, which could be based on the different methods used for oxHDL and Dys-HDL determination. Furthermore, other articles have proposed oxHDL as a biomarker for human pathologies, including nonalcoholic fatty liver disease, pericoronary inflammation, cardiovascular risk and calcific aortic valve disease [70,71,72,73]

Reaching high INDEX values is required for the concurrent generation of oxHDL and consumption of HDL. Thus, prothrombotic actions of oxHDL appear while protective HDL properties on the endothelium decrease [17,19,74], promoting the prothrombotic phenotype. Thus, it is possible to have a permanent conversion from HDL into oxHDL in SSP with a normal or high plasma HDL level upon ICU admission. Low-INDEX results from a slightly increased oxHDL level and marginally decreased or normal HDL levels. In this scenario, a competition between prothrombotic and protective actions could be generated, canceling each other out. Thus, it is possible to understand that despite the finding that low-oxHDL levels and low-INDEX values were higher than those observed in HV, the promotion of prothrombotic activity failed to occur. The plasma oxHDL level and INDEX in both SSP and the high-SSP group showed significant capabilities for predicting mortality. Remarkably, this capacity for predicting mortality was better than the widely used scores of gravity and organic failure, such as the APACHE-II and SOFA scores, respectively. These relevant findings suggest that plasma oxHDL levels and INDEX are accurate diagnostic tools for predicting mortality in SSP.

Endothelial tissue performs several crucial functions, including the control of hemostasis. However, EC becomes dysfunctional during sepsis, which is an early sign of the loss of hemostasis control in the severe systemic inflammation that triggers sepsis-associated MOD [75,76]. Endothelial dysfunction contributes to severity in dysfunctional coagulation and supports the pathogenesis of sepsis-induced coagulopathy and increasing mortality [77]. Since sepsis is a systemic inflammatory syndrome, several studies have reported that markers of inflammation are useful for diagnostic of sepsis-associated coagulopathy [78]. However, these strategies demand the evaluation of several inflammatory mediators, thus making such an evaluation unsuitable for transfer to a clinical setting. Along the same line, it has been reported that coagulation parameters are suitable biomarkers for predicting coagulopathy in septic patients [79,80], but again, several coagulation parameters need to be measured. Interestingly, ISTH-DIC also predicts mortality in non-septic patients [81].

Evidence indicates that the scavenger receptor class BI (SR-BI) is an HDL receptor that mediates its protective actions [82]. The results from this study indicate that oxHDL actions were mediated by the LOX-1 receptor. Conversely, evidence has been presented in which oxHDL interacts with SR-BI in platelets [83]. This finding suggests that HDL and oxHDL actions are mediated by different receptors, which is in agreement with different outcomes. Interestingly, LOX-1 is an LDL and oxLDL receptor [27,84], indicating that oxHDL elicits its deleterious actions because it interacts with the LDL receptor. On the other hand, oxHDL activates the nuclear factor kappa beta (NF-κB) transcription factor, a process mediated by LOX-1 activation, thus modulating protein expression [27,85]. Similar to the results reported in this study, the LOX-1 receptor mediates the oxHDL-induced expression regulation of coagulation factors and adhesion proteins. Further experiments must be performed to demonstrate whether NF-κB is involved in this transcriptional process.

Several lines of evidence indicate that endotoxin binds TLR-4 to mediate its inflammatory action in EC and other cell types [55,86,87]. This evidence suggests that endotoxin plays a key role in a dysfunctional endothelium. However, our results demonstrate that oxHDL triggers protein expression related to coagulation in the absence of TLR-4, but the complete expression of TF, TAFI and P-Sel requires the endotoxin-mediated TLR-4 activation. Interestingly, endotoxin binds to HDL, thus neutralizing the endotoxin’s properties [88], indicating that even when the plasma HDL level is maintained in a normal range, its actions could be blunted by its interaction with endotoxin.

The main strengths of our study are the monitoring of patients for 30 days or until death. The study design and experimental procedures were performed in a rigorous double-blind manner; thus, researchers performing benchtop experiments did not participate in patient management or the survival evaluation carried out by the medical personnel, and a non-survival outcome was significant in allowing successful discrimination between patients. However, our study also has some limitations, mainly regarding the capacity of the oxHDL level and INDEX to predict SSP mortality. Because we used a rigorous inclusion and exclusion criteria, patients were classified based on a restricted definition of septic shock. Thus, we used a low number of patients that is not high enough for our investigation to be considered as a clinical study or clinical trial. Additionally, we did not use a group without septic shock because this group of patients did not show any significant results. Further studies and a bigger sample size are required to extrapolate these findings to a general population of patients who are undergoing septic shock.

Taken together, the findings shown here indicate that the plasma oxHDL level and particularly its INDEX are correlated with coagulopathy in SSP, which is associated with increased mortality in SSP.

## 5. Conclusions

Increased plasma oxHDL levels and particularly the oxHDL/HDL INDEX are correlated with coagulopathy in SSP, with a severe impact on mortality and risk of death, which is mediated through an underlying molecular mechanism that involves changes in endothelial protein expression. Taken together, the findings shown here indicate that plasma oxHDL and the oxHDL/HDL INDEX levels are associated with increased mortality in SSP.

## Figures and Tables

**Figure 1 antioxidants-12-00543-f001:**
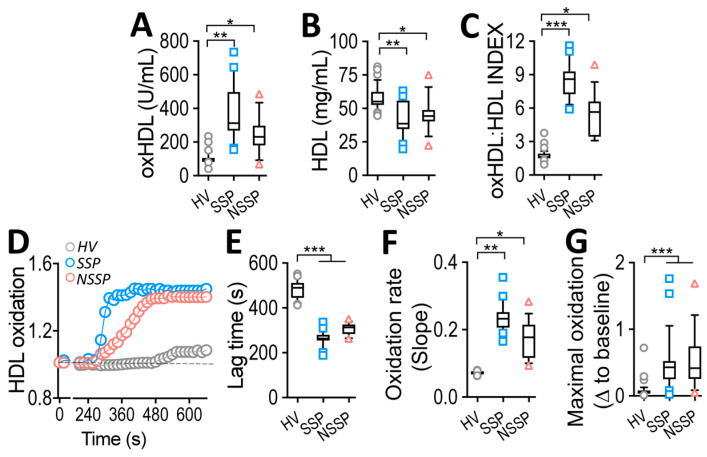
HDL and oxHDL plasma levels in SSP and NSSP and oxidation kinetic parameters. (**A**–**D**) Blood samples were collected from HV (grey circle), SSP (blue square) and NSSP (red triangle) to measure plasma levels of oxHDL (**A**), HDL (**B**) and INDEX (**C**). (**D**) Representative traces for HDL time-lapse oxidation progression from native HDL obtained from HV (grey circles), SSP (red circles) and NSSP (blue circles). (**E**–**G**) Determination of lag time (**E**), oxidation rate (**F**) and maximal oxidation (**G**) for native HDL obtained from HV (grey circles), SSP (red circles) and NSSP (blue circles). Statistical differences were assessed by a one-way analysis of variance (ANOVA) (Kruskal–Wallis) followed by Dunn’s post hoc test. * *p* < 0.05, ** *p* < 0.01, *** *p* < 0.001. Values are expressed as the median ± 10–90 percentile.

**Figure 2 antioxidants-12-00543-f002:**
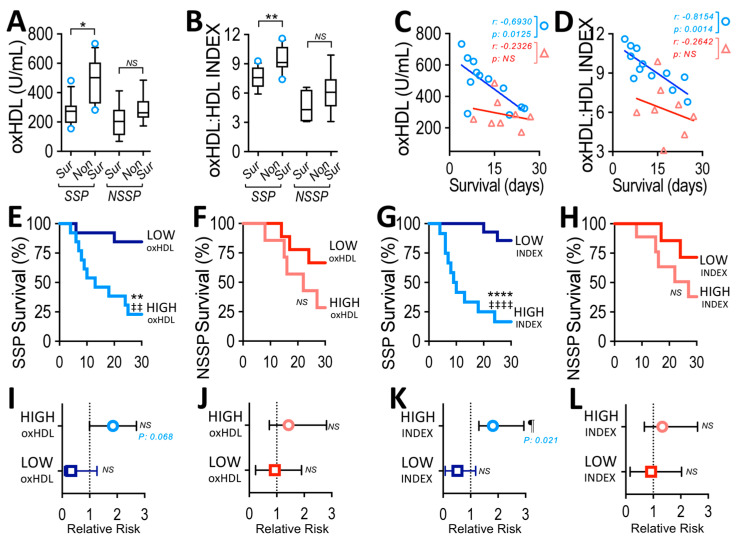
OxHDL plasma level and INDEX in surviving and non-surviving patients, survival time correlation, survival curve comparisons and relative risk of death analyses in SSP and NSSP. (**A**,**B**) Plasma level of oxHDL (**A**) and INDEX (**B**) in surviving and non-surviving SSP (blue circles) and NSSP (red triangles). Statistical differences were assessed with student’s *t*-test (Mann–Whitney) for SSP and NSSP. * *p* < 0.05, ** *p* < 0.01, NS: non-significant. Values are expressed as the median ± 10–90 percentile. (**C**,**D**) Correlation analyses between oxHDL (**B**) and INDEX (**C**) with days of permanence at ICU for non-surviving patients within a 30-day time frame in SSP (blue circle) and NSSP (red triangle) groups (*r* = −0.6930; *p* = 0.0125 and *r* = −0.2326; *p* = 0.5793, respectively, for C, and *r* = −0.8154; *p* = 0.0012 and *r* = −0.2642; *p* = 0.5271, respectively, for (**D**)). Survival (Kaplan–Meier) curves within a 30-day time frame for SSP (**E**,**G**) and NSSP (**F**,**H**) comparing high- and low-oxHDL (**E**,**F**) or comparing high- and low-INDEX (**G**,**H**). ** *p* = 0.01 (log-rank (Mantel–Cox) test) ‡‡, *p* = 0.01 (Gehan–Breslow–Wilcoxon test) when comparing high-oxHDL versus low-oxHDL groups. **** *p* < 0.0001 (log-rank (Mantel–Cox) test). ‡‡‡‡, *p* < 0.0001 (Gehan–Breslow–Wilcoxon test) when comparing high-INDEX versus low-INDEX groups. NS: non-significant. Contingency analyses performed to determine relative risk in SSP (**I**,**K**) and NSSP (**J**,**L**) comparing high- and low-oxHDL groups (**I**,**J**) or high- and low-INDEX groups (**K**,**L**). Values are expressed as the mean ± 95% CI. ¶ *p* = 0.021 (Fisher’s exact test) comparing high- versus low-oxHDL level groups within a 30-day time frame. NS: non-significant.

**Figure 3 antioxidants-12-00543-f003:**
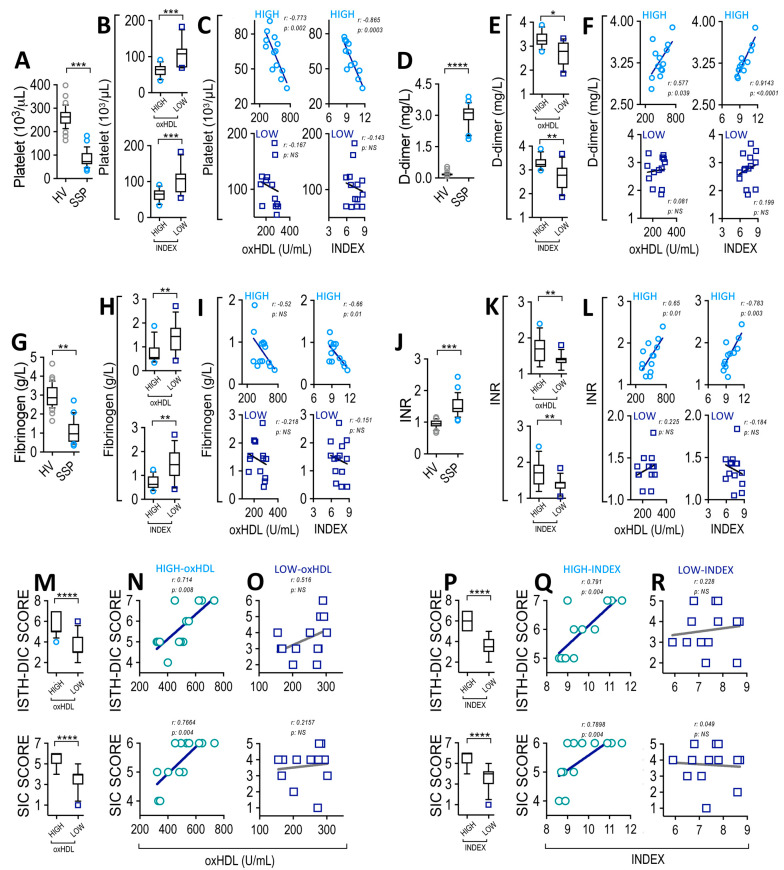
Plasma level of ISTH-DIC and SIC parameters correlates with oxHDL and INDEX in SSP. Plasma level of platelet count (**A**), D-dimer (**D**), fibrinogen (**G**), and INR (**J**) in HV and SSP. Plasma level of platelet count (**B**), D-dimer (**E**), fibrinogen (**H**), and INR (**K**) in high- and low-oxHDL and high- and low-INDEX groups from SSP. Statistical differences were assessed with student’s t-test (Mann–Whitney). * *p* < 0.05, ** *p* < 0.01, *** *p* < 0.001, **** *p* < 0.0001. Correlation analyses between: platelet count with high-oxHDL ((**C**), upper-left panel, *r* = −0.773; *p* = 0.002), low-oxHDL ((**C**), lower-left panel, *r* = −0.167; *p* = NS), high-INDEX ((**C**), upper-right panel, *r* = −0.865; *p* = 0.0003) and low-INDEX ((**C**), lower-right panel, *r* = −0.1430; *p* = NS), D-dimer with high-oxHDL ((**F**), upper-left panel, *r* = 0.577; *p* = 0.039), low-oxHDL ((**F**), lower-left panel, *r* = 0.081; *p* = NS), high-INDEX ((**F**), upper-right panel, *r* = 0.9143; *p* < 0.0001) and low-INDEX ((**F**), lower-right panel, *r* = 0.1998; *p* = NS), fibrinogen with high-oxHDL ((**I**), upper-left panel, *r* = −0.52; *p* = NS), low-oxHDL ((**I**), lower-left panel, *r* = −0.2185; *p* = NS), high-INDEX ((**I**), upper-right panel, *r* = −0.660; *p* = 0.01) and low-INDEX ((**I**), lower-right panel, *r* = −0.1518; *p* = NS), INR with high-oxHDL ((**L**), upper-left panel, *r* = 0.650; *p* = 0.01), low-oxHDL ((**L**), lower-left panel, *r* = 0.2275; *p* = NS), high-INDEX ((**L**), upper-right panel, *r* = 0.783; *p* = 0.003) and low-INDEX ((**L**), lower-right panel, *r* = −0.184; *p* = NS). (**M**–**R**) SIC score determination in high- and low-oxHDL (**M**) and high- and low-INDEX (**P**) groups from SSP. Statistical differences were assessed with student’s *t*-test (Mann–Whitney). **** *p* < 0.0001. SIC score determination in oxHDL plasma level and INDEX SSP in high and low groups. Correlation analyses betweenSIC score with high-oxHDL ((**N**), *r* = 0.714; *p* = 0.008 and *r* = 0.7664; *p* = 0.004, respectively), SIC score with low-oxHDL ((**O**), *r* = 0.516; *p* = NS and *r* = 0.2157; *p* = NS, respectively), SIC score with high-INDEX ((**Q**), *r* = 0.791; *p* = 0.004 and *r* = 0.7898; *p* = 0.004, respectively), SIC score with low-INDEX ((**R**), *r* = 0.228; *p* = NS and *r* = 0.049; *p* = NS, respectively).

**Figure 4 antioxidants-12-00543-f004:**
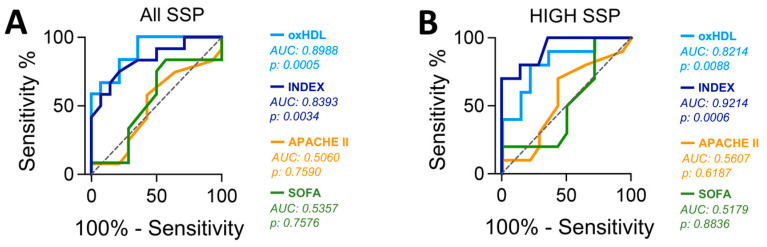
Plasma level of oxHDL and INDEX in SSP. AUROC curve analysis in all SSP (**A**) and non-surviving SSP (**B**) were performed for the plasma level of high-oxHDL ((**A**), AUC = 0.8988, *p* = 0.0005 (95% CI: 0.7835–1.001); (**B**), AUC = 0.8214, *p* = 0.0885 (95% CI: 0.6480–0.9948)), high-INDEX ((**A**), AUC = 0.8393, *p* = 0.0034 (95% CI: 0.6838–0.9948); (**B**), AUC = 0.9214, *p* = 0.0006 (95% CI: 0.8155–0.9998)), APACHE II ((**A**), AUC = 0.5060, *p* = 0.7590 (95% CI: 0.2753–0.7366); (**B**), AUC = 0.5607, *p* = 0.6187 (95% CI: 0.3218–0.7996)) and SOFA ((**A**), AUC = 0.5357, *p* = 0.7576 (95% CI: 0.3022–0.7692); (**B**), AUC = 0.5179, *p* = 0.8836 (95% CI: 0.2738–0.7619)) scores. A diagonal discontinuous line in plots is the non-discrimination line. The best sensitivity (S) and specificity (E) to predict mortality outcome in SSP and the corresponding Youden INDEX (YI) in all SSP (A) were oxHDL: S:83,33%, E:78,57%, YI:0.62, cut-off ≥312.7; INDEX: S:75.0%, E:91.11%, YI:0.54, cut-off ≥8.65%; APACHE-II: S:58.33%, E:57.14% and YI:0.15, cut-off ≥23.5; and SOFA: S:83.33%, E:42.86% and YI:0.26, cut-off ≥9.5, whereas in high-SSP (**B**) groups, they were: high-oxHDL: S:80,11%, E:71,43%, YI:0.5857, cut-off ≥338.5; high-INDEX: S:80.0%, E:85.71%, YI:0.66, cut-off ≥8.75%; high-APACHE-II: S:70.53%, E:57.14% and YI:0.27, cut-off ≥23.5; and high-SOFA: S:70.00%, E:28.57% and YI:0.29, cut-off ≥13.5.

**Figure 5 antioxidants-12-00543-f005:**
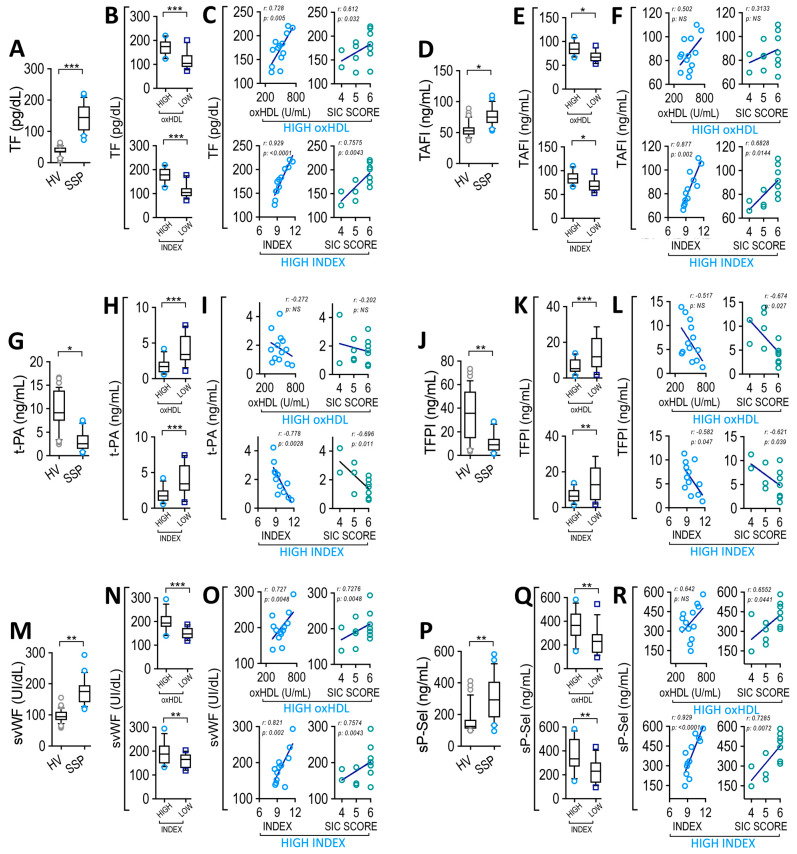
Plasma level of coagulation factors and soluble platelet adhesion proteins correlates with oxHDL and INDEX in SSP. Plasma level of TF (**A**), TAFI (**D**), t-PA (**G**), TFPI (**J**), svWF (**M**) and sP-Sel (**P**) in HV and SSP. Plasma level of TF (**B**), TAFI (**E**), t-PA (**H**), TFPI (**K**), svWF (**N**) and sP-Sel (**Q**) in high- and low-oxHDL and high- and low-INDEX groups from SSP. Statistical differences were assessed with student’s *t*-test (Mann–Whitney). * *p* < 0.05, ** *p* < 0.01, *** *p* < 0.001. Values are expressed as the median ± 10–90 percentile. Correlation analyses between: TP with high-oxHDL ((**C**), left panel, *r* = 0.728; *p* = 0.005), SIC score from high-oxHDL ((**C**), middle-left panel, *r* = 0.612; *p* = 0.032), high-INDEX ((**C**), middle-right panel, *r* = 0.929; *p* < 0.0001) and SIC score from high-INDEX ((**C**), right panel, *r* = 0.7575; *p* = 0.0043), TAFI with high-oxHDL ((**F**), left panel, *r* = 0.502; *p* = 0.08), SIC score from high-oxHDL ((**F**), middle-left panel, *r* = 0.3133; *p* = 0.297), high-INDEX ((**F**), middle-right panel, *r* = 0.877; *p* = 0.002) and SIC score from high-INDEX ((**F**), right panel, *r* = 0.6828; *p* = 0.0144), t-PA with high-oxHDL ((**I**), left panel, *r* = −0.2726; *p* = 0.3676), SIC score from high-oxHDL ((**I**), middle-left panel, *r* = −0.202; *p* = 0.509), high-INDEX ((**I**), middle-right panel, *r* = −0.7789; *p* = 0.0028) and SIC score from high-INDEX ((**I**), right panel, *r* = −0.696; *p* = 0.011), TFPI with high-oxHDL ((**L**), left panel, *r* = −0.517; *p* = 0.071), SIC score from high-oxHDL ((**L**), middle-left panel, *r* = −0.674; *p* = 0.027), high-INDEX ((**L**), middle-right panel, *r* = −0.5823; *p* = 0.047) and SIC score from high-INDEX (L, right panel, *r* = −0.621; *p* = 0.039), svWF with high-oxHDL ((**O**), left panel, *r* = 0.7276; *p* = 0.0048), SIC score from high-oxHDL ((**O**), middle-left panel, *r* = 0.7276; *p* = 0.0048), high-INDEX ((**O**), middle-right panel, *r* = 0.821; *p* = 0.002) and SIC score from high-INDEX ((**O**), right panel, *r* = 0.7475; *p* = 0.0043), sP-Sel with high-oxHDL ((**R**), left panel, *r* = 0.6429; *p* = 0.1389), SIC score from high-oxHDL ((**R**), middle-left panel, *r* = 0.6552; *p* = 0.0441), high-INDEX ((**R**), middle-right panel, *r* = 0.929; *p* < 0.0001) and SIC score from high-INDEX ((**R**), right panel, *r* = 0.7285; *p* = 0.0072).

**Figure 6 antioxidants-12-00543-f006:**
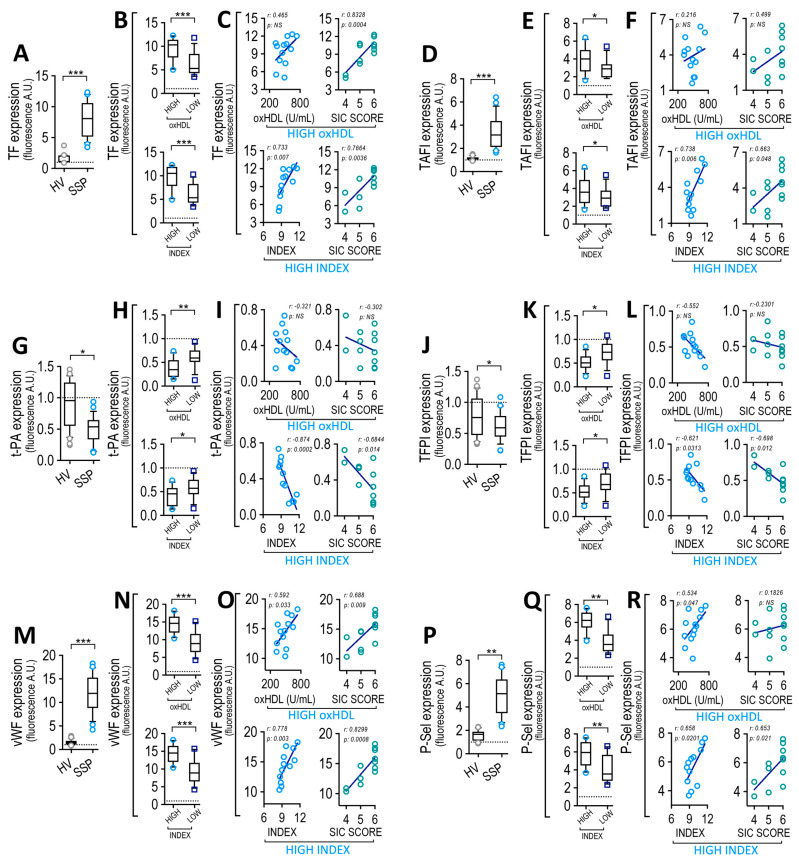
Coagulation factors and soluble platelet adhesion protein expression in CMEC from SSP. Normalized expression level of TF (**A**), TAFI (**D**), t-PA (**G**), TFPI (**J**), svWF (**M**) and sP-Sel (**P**) in HV and SSP. Plasma level of TF (**B**), TAFI (**E**), t-PA (**H**), TFPI (**K**), svWF (**N**) and sP-Sel (**Q**) in high- and low-oxHDL and high- and low-INDEX groups from SSP. Statistical differences were assessed with student’s t-test (Mann–Whitney). * *p* < 0.05, ** *p* < 0.01, *** *p* < 0.001. Values are expressed as the median ± 10–90 percentile. Correlation analyses between: TP with high-oxHDL ((**C**), left panel, *r* = 0.465; *p* = 0.109), SIC score from high-oxHDL ((**C**), middle-left panel, *r* = 0.8328; *p* = 0.0004), high-INDEX ((**C**), middle-right panel, *r* = 0.733; *p* = 0.007) and SIC score from high-INDEX ((**C**), right panel, *r* = 0.7664; *p* = 0.0036), TAFI with high-oxHDL ((**F**), left panel, *r* = 0.2168; *p* = 0.477), SIC score from high-oxHDL ((**F**), middle-left panel, *r* = 0.499; *p* = 0.167), high-INDEX (F, middle-right panel, *r* = 0.738; *p* = 0.006) and SIC score from high-INDEX ((**F**), right panel, *r* = 0.663; *p* = 0.048), t-PA with high-oxHDL ((**I**), left panel, *r* = −0.3213; *p* = 0.2845), SIC score from high-oxHDL ((**I**), middle-left panel, *r* = −0.302; *p* = 0.3169), high-INDEX ((**I**), middle-right panel, *r* = −0.8743; *p* = 0.0002) and SIC score from high-INDEX (I, right panel, *r* = −0.6844; *p* = 0.03169), TFPI with high-oxHDL ((**L**), left panel, *r* = −0.552; *p* = 0.0505), SIC score from high-oxHDL ((**L**), middle-left panel, *r* = −0.2301; *p* = 0.4495), high-INDEX ((**L**), middle-right panel, *r* = −0.621; *p* = 0.0313) and SIC score from high-INDEX ((**L**), right panel, *r* = −0.698; *p* = 0.012), vWF with high-oxHDL ((**O**), left panel, *r* = 0.592; *p* = 0.033), SIC score from high-oxHDL ((**O**), middle-left panel, *r* = 0.688; *p* = 0.009), high-INDEX ((**O**), middle-right panel, *r* = 0.778; *p* = 0.003) and SIC scores from high-INDEX ((**O**), right panel, *r* = 0.8299; *p* = 0.0008), P-Sel with high-oxHDL ((**R**), left panel, *r* = 0.5344; *p* = 0.047), SIC score from high-oxHDL ((**R**), middle-left panel, *r* = 0.1826; *p* = 0.5504), high-INDEX ((**R**), middle-right panel, *r* = 0.658; *p* = 0.0201) and SIC score from high-INDEX ((**R**), right panel, *r* = 0.653; *p* = 0.021). Dotted line depicts normalized expression obtained in non-treated EC cultures.

**Figure 7 antioxidants-12-00543-f007:**
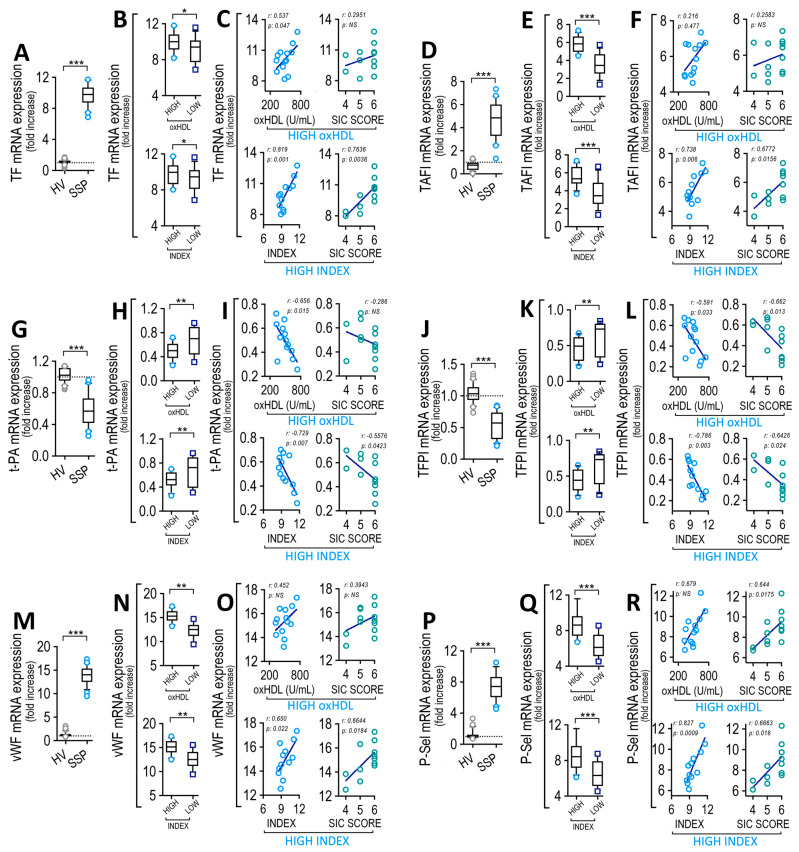
Coagulation factors and platelet adhesion proteins expression in cultured EC exposed to plasma from SSP. Plasma samples were collected from HV, high-oxHDL and low-oxHDL SSP blood samples and then added to cultured EC for 24 h. Normalized mRNA expression of TF (**A**), TAFI (**D**), t-PA (**G**), TFPI (**J**), svWF (**M**) and sP-Sel (**P**) in HV and SSP. Plasma level of TF (**B**), TAFI (**E**), t-PA (**H**), TFPI (**K**), svWF (**N**) and sP-Sel (**Q**) in high- and low-oxHDL and high- and low-INDEX groups from SSP. Statistical differences were assessed with student’s t-test (Mann–Whitney). * *p* < 0.05, ** *p* < 0.01, *** *p* < 0.001. Values are expressed as the median ± 10–90 percentile. Correlation analyses between: TP with high-oxHDL ((**C**), left panel, r = 0.537; *p* = 0.047), SIC score from high-oxHDL ((**C**), middle-left panel, r = 0.2951; *p* = 0.3277), high-INDEX ((**C**), middle-right panel, r = 0.819; *p* = 0.001) and SIC score from high-INDEX ((**C**), right panel, r = 0.7636; *p* = 0.0038), TAFI with high-oxHDL ((**F**), left panel, r = 0.2168; *p* = 0.477), SIC score from high-oxHDL ((**F**), middle-left panel, r = 0.2583; *p* = 0.3943), high-INDEX ((**F**), middle-right panel, r = 0.738; *p* = 0.006) and SIC score from high-INDEX (F, right panel, r = 0.6772; *p* = 0.0156), t-PA with high-oxHDL ((**I**), left panel, r = −0.656; *p* = 0.015), SIC score from high-oxHDL ((**I**), middle-left panel, r = −0.286; *p* = 0.03434), high-INDEX ((**I**), middle-right panel, r = −0.729; *p* = 0.007) and SIC score from high-INDEX ((**I**), right panel, r = −0.5576; *p* = 0.0423), TFPI with high-oxHDL ((**L**), left panel, r = −0.591; *p* = 0.0335), SIC scores from high-oxHDL ((**L**), middle-left panel, r = −0.662; *p* = 0.013), high-INDEX ((**L**), middle-right panel, r = −0.786; *p* = 0.003) and SIC score from high-INDEX ((**L**), right panel, r = −0.6426; *p* = 0.024), vWF with high-oxHDL ((**O**), left panel, r = 0.4524; *p* = 0.1206), SIC score from high-oxHDL ((**O**), middle-left panel, r = 0.3943; *p* = 0.1825), high-INDEX ((**O**), middle-right panel, r = 0.650; *p* = 0.022) and SIC score from high-INDEX ((**O**), right panel, r = 0.6644; *p* = 0.0184), P-Sel with high-oxHDL ((**R**), left panel, r = 0.6799; *p* = 0.0106), SIC score from high-oxHDL ((**R**), middle-left panel, r = 0.644; *p* = 0.0175), high-INDEX (R, middle-right panel, r = 0.827; *p* = 0.0009) and SIC score from high-INDEX ((**R**), right panel, r = 0.6663; *p* = 0.018). Dotted line depicts normalized expression obtained in non-treated EC cultures.

**Figure 8 antioxidants-12-00543-f008:**
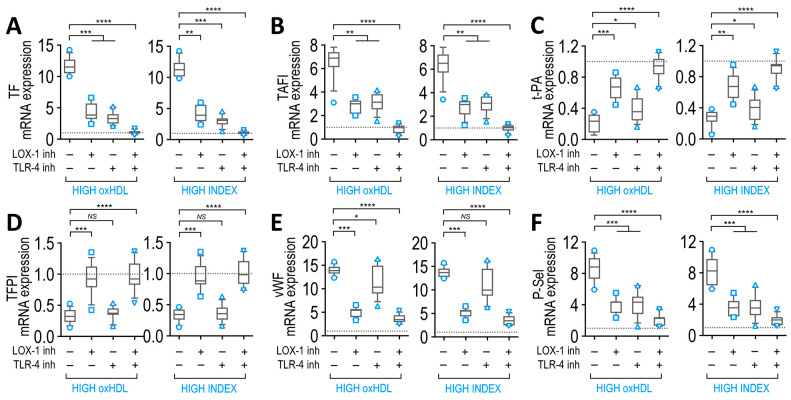
Effects of LOX-1 and TLR-4 recepto inhibitors in coagulation factors and platelet adhesion protein mRNA expression in cultured EC exposed to plasma from SSP. Plasma samples were collected from high-oxHDL and high-INDEX SSP blood samples and then added to cultured EC for 24 h, in the absence (−) or presence (+) of the neutralizing antibody against LOX-1 function (LOX-1 inh) and the TLR-4-specific inhibitor CLI-095 (TLR-4 inh), and then mRNA expression of coagulation factors and platelet adhesion proteins were measured. Normalized mRNA expression of TF (**A**), TAFI (**B**), t-PA (**C**), TFPI (**D**), vWF (**E**) and P-Sel (**F**) were measured in EC cultures exposed to high-oxHDL (left panels) and high-INDEX (right panels) plasma samples. Statistical differences were assessed by a one-way analysis of variance (ANOVA) (Kruskal–Wallis) followed by Dunn’s post hoc test. * *p* < 0.05, ** *p* < 0.01, *** *p* < 0.001, **** *p* < 0.0001. Values are expressed as the median ± 10–90 percentile. NS: non-significant. Dotted line depicts normalized expression obtained in non-treated EC cultures.

**Figure 9 antioxidants-12-00543-f009:**
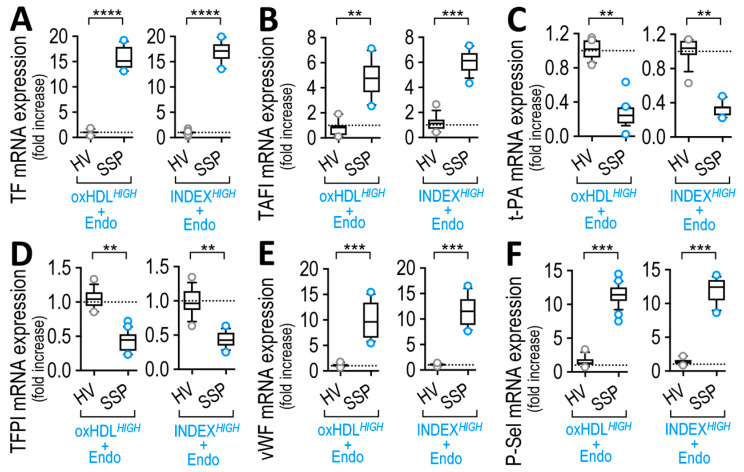
Coagulation factors and platelet adhesion protein mRNA expression in cultured EC exposed to exogenous SSP plasma preparation. Exogenous plasma preparation of SSP high- and low-oxHDL were designed using the median concentration of oxHDL measured in plasma obtained from high- and low-oxHDL groups (SSP-oxHDL^HIGH^+Endo and SSP-oxHDL^LOW^+Endo). Exogenous plasma preparation of SSP high- and low-INDEX were designed using the concentration of oxHDL and HDL measured in the median INDEX value determined in plasma obtained from high- and low-INDEX groups (SSP-INDEX^HIGH^+Endo and SSP-INDEX^LOW^+Endo). Exogenous plasma preparation from HV was designed combining a proper amount of HDL and oxHDL but without endotoxin (HV-oxHDL and HV-INDEX). Normalized mRNA expression of TF (**A**), TAFI (**B**), t-PA (**C**), TFPI (**D**), vWF (**E**) and P-Sel (**F**) were measured in EC cultures exposed to SSP-oxHDL^HIGH^+Endo (left panels) and SSP-INDEX^HIGH^+Endo (right panels) exogenous SSP plasma preparation. Statistical differences were assessed with student’s t-test (Mann–Whitney). ** *p* < 0.01, *** *p* < 0.001, **** *p* < 0.0001 Values are expressed as the median ± 10–90 percentile. Dotted line depicts normalized expression obtained in non-treated EC cultures.

## Data Availability

The data presented in this study are available in the supplementary material or on suitable request from the corresponding authors.

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
