# Peer review of "Sepsis-Induced Coagulopathy Phenotype Induced by Oxidized High-Density Lipoprotein Associated with Increased Mortality in Septic-Shock Patients"

_antioxidants, 2023, doi:10.3390/antiox12030543_

Round 1
Reviewer 1 Report
This is an interesting addition to the literature, with a focus on factors that block coagulation that occurs during sepsis in humans. It also describes how some of these protective factors are neutralized by products of sepsis. The report has a focus on HDL which is oxidized to oxHDL and promotes hemostatic dysfunction. There is a multitude of data which is useful. The authors also describe how the sepsis factors cause dysfunction of vascular endothelial and epithelial cells. In general, this is a useful report that should be useful to caretakers treating septic patients.
This report relates to an important area in the field of sepsis, namely the strong prothrombotic series of events that can lead to structural damage of much of the clotting system. While it is very well established that the coagulation system is activated during sepsis, the details described in this report are unique: HDL undergoes oxidation to oxHDL, which results in hemostatic dysfunction, acquiring prothrombotic activities. Such information is new and represents complex events which need to be considered by clinicians.
Reviewer 2 Report
The authors tried to demonstrate that oxLDL associated with DIC and mortality in septic shock activate LOX-1 and TLR4 to create a pro-coagulatory environment by increasing mRNA expression of TF, TAFI, vWF, and p-selectin and reducing t-PA and TFPI. However, the clinical study is not convincing due to lack of severe sepsis without shock. Since sepsis is a heterogeneous diseases, the sample sizes seems too small to justify a biomarker. The zebra fish model is too far from human and the data obtain is not convincing. In addition, no clear evidence demonstrate the pathways of oxLDL-LOX1/TLR-4-factors that affect coagulation. More importantly, these factors are not necessarily related to DIC.
The terms of SSP is very confusing in the manuscript, some represents septic-shock patients, whilst others represent shock-septic patients.
All the figures are too big and non-professional.
Overall, there are not sufficient and solid evidence to support the conclusion
Reviewer 3 Report
This is a well-written and interesting manuscript and I have very few objections. However, please consider using the term "sepsis-induced coagulopathy" (PMID: 34344558; DOI: 10.1016/j.arcmed.2021.07.003), instead of "Disseminated Intravascular coagulation".
If "Disseminated Intravascular coagulation" is preferred to "sepsis-induced coagulopathy", please explain the rationale.
Several abbreviations are not explained the first time they appear in the main text (e.g. in 2.3). ICU or UCI? Please, be uniform. ICU is more conventional.
